# Schiff Bases Derived from Pyridoxal 5′-Phosphate and 2-X-Phenylamine (X = H, OH, SH): Substituent Effects on UV-Vis Spectra and Hydrolysis Kinetics

**DOI:** 10.3390/molecules29153504

**Published:** 2024-07-26

**Authors:** Maksim N. Zavalishin, Aleksei N. Kiselev, George A. Gamov

**Affiliations:** 1Department of General Chemical Technology, Ivanovo State University of Chemistry and Technology, Sheremetevskii Pr. 7, Ivanovo 153000, Russia; zavalishin00@gmail.com; 2G.A. Krestov Institute of Solution Chemistry, Russian Academy of Science, Akademicheskaya Str. 1, Ivanovo 153045, Russia; scatol@yandex.ru

**Keywords:** Schiff base, hydrolysis, rate constant, molecular orbitals, molecular electron density theory (MEDT)

## Abstract

Schiff bases are compounds that are widely distributed in nature and have practical value for industry and biomedicine. Another important use of Schiff bases is identifying metal ions and different molecules, including proteins. Their proneness to hydrolysis limits the utilization of Schiff bases to mainly non-aqueous solutions. However, by introducing –OH and –SH substituents to aromatic amine-bearing rings, it is possible to increase the resilience of the Schiff base to destruction in water. The present paper discusses how the hydroxyl or thiol group influences the spectral properties and kinetics of the hydrolysis and formation of Schiff bases derived from pyridoxal 5′-phosphate and aniline, 2-hydroxyaniline, and 2-mercaptoaniline using quantum chemical data. The spectral variation between different imines can be explained by taking into account the geometry and frontier molecular orbital alteration induced by the substituents. The changes in the hydrolysis rate are analyzed using the computed values of local reactivity indices.

## 1. Introduction

Schiff bases (SB) are versatile compounds that play an important role in nature. For example, they make a binding chromophore during the process of chloride transport through biological membranes [1]. SB counter-ions can be also found in light-driven sodium pumps, where they mediate the function of a protein [2]. In human practice, Schiff bases are widely used for producing catalysts [3,4,5,6], promising sorbents, and separating materials [7,8] for the recovery of valuable metals [9], removal of toxic elements [10], and desalination [11,12]. Polyimines (polymeric Schiff bases) are known for their low electrical resistance [13], which makes them perfect for developing conducting composites. The spectrum of SB utilized for medical purposes is also broad. These compounds can be of use on their own as antibacterial [14,15], antiviral (including SARS-Cov-2 [16]), antifungal [15], antitumor [15,17], anticoagulant [18], and antidiabetic [19] agents. However, SBs can also make a valuable addition to composite biomedical applications, including novel bandages that accelerate the healing of wounds [20], blends designed for the treatment of delicate organs such as eyes [21], magnetic nanoparticles with antifungal activity [22], biogels with controlled release of bioactive-compounds [23], or those suitable for 3D bioprinting [24]. Complexing with metal ions is arguably the most frequent method of enhancing the biological activity of the starting SB [25,26,27].

In addition, the great chelating ability of Schiff bases in regard to different metal ions is utilized for metal recognition (see, e.g., [28,29]), as the complex formation alters the electronic structure of the molecules, which results in the alteration of the UV-Vis spectrum and gain (or loss) of fluorescence. SBs are also often believed to identify anions in solutions [30,31,32]; however, it should be kept in mind that the anion recognition mechanism involves proton exchange between the Schiff base and an anion serving as a base [33]. Therefore, anion sensors based on SBs are rather pH-sensitive indicators in non-aqueous solvents.

Aside from benefits such as versatility, ease of synthesis and tuning of the SB molecule for a specific purpose, several disadvantages of the Schiff bases are known. Their proneness to hydrolysis in an aqueous medium is probably the most important as water is the greenest possible solvent with the most practical significance [34]. To overcome this problem, non-aqueous solvents are often necessary to prevent the Schiff bases from water-induced destruction. However, when studying biological objects such as DNA or protein molecules, sometimes it is impossible to avoid a water-rich medium.

In this regard, two Schiff bases have drawn our attention: the derivatives of pyridoxal 5′-phosphate (PLP) and either 2-hydroxyphenylamine (**2**) (See Figure 1 for structures) [35] or 2-mercaptophenylamine (**3**) [36], which were further applied for identifying albumins [37,38] in aqueous medium. It happened that we recently studied the formation and hydrolysis of a similar Schiff base (**1**) derived from PLP and aniline without any substituent at site 2 [39], and found it to hydrolyze readily in water at different pH values (4.5, 7.4, 9.5). However, for both hydroxyl and thiol derivatives of SB, no observable hydrolysis was reported in the literature [35,36,37,38]. That is why we decided to study the hydrolysis of SBs **2** and **3** and determine if OH- and SH- groups introduced into the phenyl ring increase the stability of the imine. However, first we studied the substituent effect on the UV-Vis spectra of SBs, which is of additional interest to us.

## 2. Results and Discussion

### 2.1. Spectral Properties of SBs in DMSO

SBs in the solid state differ in color: **1** is orange, **2** is reddish, and **3** is slightly yellow. The UV-Vis spectra of various Schiff bases recorded in DMSO are also different (Figure 2).

The introduction of the –OH group induces a bathochromic shift in the spectrum compared to that of unsubstituted SB **1**, while the mercapto group leads to a blue shift in the spectrum. The results of quantum chemical calculations show that the spectral changes caused by the substituents are the consequence of two factors. The first one is the dihedral angle between two aromatic rings that determines the efficiency of π-π conjugation. The other one is the involvement of oxygen or sulfur atoms of the substituent in frontier molecular orbitals. For example, when the first conformers of each SB are analyzed (see the structures of conformers and the related xyz coordinates of atoms in Appendix A), the introduction of –OH leads to a negligible change in the dihedral angle (from 38.2° to 39.6°) and a relatively strong contribution of oxygen to the highest occupied (HOMO) and lowest unoccupied molecular orbitals (LUMO) (HOMO and LUMO are given in Appendix A). However, the thiol group increases the dihedral angle to 44.5° (which decreases significantly the contribution from the pyridine ring to HOMO, see Appendix A), while the sulfur influence on LUMO is almost negligible. A combination of the above reasons may explain why OH-substituted SB **2** light absorption is red-shifted (unchanged dihedral angle + OH- contribution), while the UV-Vis spectrum of SH-bearing SB **3** shows a hypsochromic shift relative to the spectrum of unsubstituted SB **1** (increased dihedral angle + less involved SH-group).

It is noteworthy that SBs can exist as different conformers arising from the rotation of either a PLP residue or 2-X-phenyl ring. However, unlike hydrazones derived from PLP, which are very labile and form many different rotation conformers [40], the diversity of stable structures for SBs under study is limited. The first conformer for every Schiff base is strongly stabilized by the intramolecular hydrogen bond between the 3-OH group of the PLP moiety and imine nitrogen, which is reflected as a global minimum on the potential energy surface (see the scanning of dihedral angle C_3(PLP)_-C_4(PLP)_-CH=N in Appendix A). Rotation of the PLP residue leads to another conformer with a higher total energy (Appendix A) and a blue-shifted UV-Vis spectrum relative to that of the first structure. For SB **1**, the third conformer is possible if the intramolecular H-bond is broken. However, it is even less preferred energetically than the second one. The potential energy surface for the rotation of either a phenyl or 2X-phenyl moiety has two global minima (Appendix A) corresponding to two equivalent structures, which applies for both possible alignments of the PLP residue. From any starting configuration tested, SBs **2** and **3** return to either conformer 1 if the intramolecular H-bond was set, or conformer 2 if the intramolecular H-bond was initially broken. That is why only two conformers for substituted SBs are available. The conformers of SBs **1** and **2** that are not stabilized by the H-bond have a higher value of the dihedral angle between aromatic systems and, as a consequence, less π-π conjugation. This explains why the UV-Vis spectra of the structures lacking the H-bond are blue-shifted in relation to those of the conformers stabilized by the H-bond.

The effect of hydrogen bond breaking was additionally studied via substituting –OH groups in SB **2** by –OCH_3_ groups (Appendix A). The exchange of the hydroxyl by the methoxy group in 2-hydroxyaniline, which –OH is only slightly involved in H-bonding with imine nitrogen, leads to a negligible change in the UV-Vis spectrum (Appendix A). However, substituting the 3-OH group of the PLP moiety with an –OCH_3_ group results in a clear blue shift in the UV-Vis spectra (Appendix A). The main reason for this hypsochromic shift is the twisting of the SB molecule (dihedral angle C_3(PLP)_-C_4(PLP)_-CH=N is 36.5° while that of SB **2** is 6.3°) as the bulky methyl group causes steric hindrance. Therefore, the conjugation between aromatic rings decreases.

Moreover, we briefly studied the effect of –OH and –SH groups introduced in the *para*-position of aniline instead of the *ortho*-position using quantum chemistry calculations (Appendix A). The UV-Vis spectrum of the *para*-OH Schiff base is blue-shifted relative to that of SB **2** (Appendix A) because LUMO involves the phenyl ring to a lesser degree when an –OH group is introduced in position 4 of the hydroxyaniline residue (Appendix A). However, the spectra of 2-SH- and 4-SH-substituted SBs are almost identical (Appendix A), which can be explained by the combination of two factors. First, the *para*-substitution creates no steric repulsion between –SH and –OH groups in a Schiff base, which leads to the smaller dihedral angle value between aromatic rings and more efficient conjugation. Second, the sulfur atom in the 4-position is completely excluded from LUMO, while the –SH in the *ortho*-position provides some contribution to LUMO.

The calculated UV-Vis spectra of the first conformers are in satisfactory agreement with the experimental spectra (Figure 2; discrepancy observed for the thiol-substituted SB can be referred to as the heavy atom effect), and they are preferred in terms of total energy. Thus, they were further used for estimating the reactivity of SBs.

The most important vertical electron transitions as well as oscillator strengths are collected in Table 1, while the form of the molecular orbitals involved in electron transitions can be found in Appendix A.

The Table 1 data show that the long-wavelength maximum of the SB light absorbance is determined by the HOMO→LUMO electron transition. Both frontier molecular orbitals involve the combined π-electron system of the Schiff base, indicating the π-π character of the band in UV-Vis spectra. No n-π transitions were observed after introducing the –OH or –SH functional group. When either a conformational change or thiol substitution decreases the conjugation degree, it slightly diminishes the contribution of the HOMO→LUMO transition to the A_1_ excited state and allows the lower occupied molecular orbitals to participate in an electron transition.

No LUMO + *n* orbitals are involved in the electron transitions, as even LUMO + 1 and LUMO + 2 are 1.7–2.0 eV higher in energy than LUMO, and are contributed mainly by the phosphate group of the PLP residue.

The difference between the fluorescent properties of the SBs under study is even more striking (Figure 3 and Appendix A). Introducing both –OH and –SH groups increase the intensity of the luminescence of a Schiff base in DMSO. However, the substituted Schiff bases are excited by light at different wavelengths to yield a relatively bright fluorescence at λ_em_ ~470 nm (Appendix A): SB **2** requires the value of λ_ex_ = 390 nm, while the thiol derivative emits best when being irradiated at 320 nm (Appendix A). These findings are in perfect agreement with the UV-Vis spectra of SBs reported (Figure 2).

Therefore, the differences in structure and electron properties of SBs bearing various substituents are significant. The hydrolysis resilience of different SBs are evaluated in the following section.

### 2.2. Hydrolysis of SBs in Water at Different pH Values

The hydrolysis of Schiff base **1** was studied in detail in our previous paper [39], while water-induced destruction of SBs **2** and **3** is investigated in the present contribution. Examples of primary experimental data can be found in Figure 4 and Appendix A.

The hydrolysis (*k*_−1_) and formation (*k*_1_) rate constants were calculated from the data for SBs **2** and **3**. The ratio of *k*_1_ to *k*_−1_ gives the conditional equilibrium constant (*K*) of Schiff base formation at a given pH value. These results are given in Table 2.

Both primary experimental kinetic data and the rate constants derived indicate that SB **2** is more stable in an aqueous solution than the Schiff base derived from PLP and aniline, as the formation equilibrium constants are 1 to 2 log units higher than the respective values known for SB **1**. However, the derivative of 2-mercaptoanilie shows great resilience to hydrolysis, especially in an acidic medium. To understand why the SB’s proneness to hydrolysis decreases according to row **1** < **2** < **3**, we estimated the global electronic properties of the Schiff bases and local parameters of atoms susceptible to nucleophilic or electrophilic attack during hydrolysis. It is widely accepted that both the direct and reverse reaction of the Schiff base formation goes through the unstable semiproduct carbinolimine [41] with the general reaction presented in Figure 5.

Therefore, the reaction centers involved in hydrolysis are methine carbon (marked with green in Figure 1) and imine nitrogen (marked with blue in Figure 1). We performed quantum chemical calculations to evaluate such parameters as electronic chemical potential (*µ*), chemical hardness (*η*), global electrophilicity (*ω*), and nucleophilicity (*N*) of non-ionized SB molecules, and the local electrophilicity (*ω_k_*) and nucleophilicity (*N_k_*), as well as the local reactivity difference *R_k_* indices of the –CH= carbon and =N– nitrogen (see Section 3.3 for details). For every SB, only the most stable conformers (1) were considered. The results are given in Table 3.

For the neutral Schiff bases molecules, calculations show a partial positive Mulliken charge on the imine nitrogen and a negative Mulliken charge on methine carbon. Local electrophilicity and nucleophilicity indices, as well as *R_k_*, also indicate that nitrogen is an attractor for electrophiles, while –CH= carbon should be classified according to L.R. Domingo [42] as an ambiphilic center, which can be attacked by both electrophiles and nucleophiles.

The most interesting finding to us is a strong correlation that is observed at any studied pH value between the value of the local reactivity difference index *R_N_* and rate constant of hydrolysis for SBs **1**–**3** (Appendix A). It could be concluded that the Schiff base derived from pyridoxal 5′-phosphate and 2-mercaptoaniline might be so resilient to destruction in water because of the low electrophilicity of imine nitrogen. In turn, this can indicate that some intermediate and rate-limiting stages of hydrolysis might be a reaction between nitrogen and hydroxide ions. However, the fastest hydrolysis rates observed in an alkaline medium (Table 2) might be instead a consequence of the ionization of SBs **2** and **3** than an abundance of attacking OH^−^ ions.

We would like to note that the increased resilience to hydrolysis does not make SBs **2** and **3** indestructible in an aqueous medium, especially one containing albumins. These proteins are capable of binding free pyridoxal 5′-phoshate, which is released during the hydrolysis of SBs, forming a Schiff base through a lysine amino acid residue [43,44,45,46]. The binding constant of PLP to albumins is relatively high (log K > 6) [47,48]. Thus, the competing reaction between PLP and albumin inevitably shifts the hydrolysis of the Schiff base to the free aldehyde and amine. This may negatively affect the efficiency and applicability of the albumin indicators proposed in the literature [37,38].

Interestingly, the failure of the Schiff bases derived from pyridoxal (PL) and 2-hydroxyaniline and SB **1** to identify albumins [38] makes sense. From our experience, pyridoxal forms more unstable Schiff bases than the phosphorylated form of B_6_ vitamin, and the equilibrium of PL-derived SBs and hydrazones formation is shifted left, while SB **1** is also a subject of intense hydrolysis [39]. Therefore, both SBs may not live long enough in an aqueous solution to detect the proteins.

## 3. Materials and Methods

### 3.1. Chemicals

PLP monohydrate (abcr GmbH, Karlsruhe, Germany) with purity of >98 % wt. was used without additional purification. Aniline from Lenreaktiv (Saint Petersburg, Russia) was distilled under vacuum with fraction selection. In addition, it was filtered using a small column filled with aluminum oxide with second-degree activity according to Brockman. 2-Hydroxyaniline from Rearus (Moscow, Russia) was purified by recrystallization from ethyl alcohol. 2-Mercaptoaniline from Merck (Darmstadt, Germany) was distilled under vacuum before use. To prepare the buffer solutions with pH values of 4.5, 7.0, and 9.5, standards of potassium hydrophthalate, potassium dihydrophosphate and sodium hydrophosphate, and sodium tetraborate, respectively, manufactured by Uralkhiminvest (Ekaterinburg, Russia) were employed. Final pH values were adjusted with HCl or NaOH standardized. DMSO (EKOS-1, Staraya Kupavna, Russia) and bidistilled water (κ = 3.6 μS cm^−1^, pH = 6.6) were used to prepare the solutions.

The synthesis of **1** is described elsewhere [39]. To obtain SBs **2** and **3**, we followed the protocols proposed in previous studies [35,36]. ^1^H NMR spectra of **2** and **3** (Appendix A) were in perfect agreement with the data reported previously [35,36] and confirmed the successful synthesis of the SBs under study.

### 3.2. Experimental Techniques

NMR spectra were registered using the Bruker Avance III 500 NMR spectrometer (Bremen, Germany) with ^1^H and ^13^C operating frequencies of 500.17 and 125.77 MHz, respectively. Temperature control (298 K) was achieved using a Bruker variable temperature unit (BVT-2000). Chemical shifts were determined relative to the external standard, HMDSO (Sigma Aldrich, Darmstadt, Germany), with an error of ±0.01 ppm for ^1^H NMR spectra and ±0.1 ppm for ^13^C NMR. The standard pulse sequence [49] from TopSpin 3.6.1 software was used to record ^1^H and ^13^C NMR spectra.

The kinetic experiments were performed analogously to ones described previously [39] using the double-beam Shimadzu UV1800 (Shimadzu, Long Beach, CA, USA) spectrophotometer in the wavelength range of 200–500 nm (310–500 nm when potassium hydrogen phthalate was used) and absorbance range of 0-1. Standard quartz cells with an optical path of 1.00 cm were used. An aliquot (10 μL) of a 0.0271 mol L^−1^ Schiff base solution in DMSO (in which SBs are stable) was quickly added to 2.7 mL of an aqueous solution buffered at the required pH value, and either set of UV-Vis spectra or absorbance at the chosen wavelength were recorded (Figure 4 and Appendix A). Primary kinetic data were processed using KINET 0.8 software [50] (accessed 18 June 2024) to yield the effective rate constants of SB hydrolysis and formation reactions.

The 1D fluorescent spectra (Figure 3) of 10^−5^ mol L^−1^ SB solutions in DMSO were recorded using an RF6000 fluorimeter (Shimadzu, USA) at the excitation wavelength λ_ex_ = 365 nm in the emission wavelength range of 380-700 nm. 2D (“heat map”) fluorescent spectra (Appendix A) were registered using excitation wavelength intervals of 300 to 450 nm and emission wavelength of 330 to 700 nm. The excitation and emission slit widths were set at 5 nm. Quartz cells with an optical path of 1.00 cm were used.

### 3.3. Computation Details

The calculations of different conformers of SBs **1**, **2**, **3** were carried out for the singlet electronic state using the Gaussian 09W program package [51]. The equilibrium geometrical parameters and normal mode frequencies (Appendix A) were calculated using the hybrid DFT computational method B3LYP [52]. The energies of vertical electronic transitions and oscillator strengths (UV-Vis spectra) were calculated using the TDDFT [53] method on the same B3LYP level. Calculated UV-Vis spectra were obtained using Lorentzian curves with a half-width of the half-maximum of 30. The 6-311G++(d,p) basis set was used to describe all atoms of the SBs under study (H, C, N, O, S, P). To take into account the solvent effect of DMSO, all calculations were performed using the polarizable continuum model (PCM) [54]. The visualization of ball-and-stick models and molecular orbitals was performed using the ChemCraft program [55].

Global electronic properties of the Schiff bases were estimated following a procedure described elsewhere [56,57] according to the Molecular Electron Density Theory (MEDT) by L.R. Domingo. The electronic chemical potential (*µ*) and chemical hardness (*η*) were computed using the energy values of the frontier molecular orbitals:*µ* ≈ 0.5(*E_HOMO_* + *E_LUMO_*)(1)
*η* ≈ *E_LUMO_* − *E_HOMO_*(2)
where *HOMO* refers to the energy of the highest occupied molecular orbital, while *LUMO* refers to that of the lowest unoccupied molecular orbital.

These values allow for calculating the global electrophilicity (*ω*) and nucleophilicity (*N*):(3)$$\omega =\frac{\mu^{2}}{2\eta }$$
*N* = *E_HOMO_* − *E_HOMO_*
_(*tetracyanoethylene*)_(4)

The local electrophilicity (*ω_k_*) and nucleophilicity (*N_k_*) indices of the *k^th^* atom in the molecule require the values of Parr’s function (*P_+k_* or *P_−k_*) at the chosen atom:*ω_k_* = *ωP*_+*k*_(5)
*N_k_* = *NP*_−*k*_(6)

Finally, the local reactivity difference index *R_k_* [42] can be calculated as follows:

If (1 < *ω_k_*/*N_k_* < 2) or (1 < *N_k_*/*ω_k_* < 2) then:*R_k_* ≈ 0.5(*ω_k_* + *N_k_*)(7)
meaning that the reaction center is ambiphilic. Otherwise,
*R_k_* ≈ *ω_k_* − *N_k_*(8)
where *R_k_* > 0 corresponds to an electrophilic center and *R_k_* < 0 corresponds to a nucleophilic center.

If |*R_k_*| < 0.10, then *R_k_* = 0.00.

The higher the value of *R_k_* is, the more pronounced the local electrophilicity or nucleophilicity is.

## 4. Conclusions

The spectral properties and the kinetics of hydrolysis and formation of the Schiff bases derived from pyridoxal 5′-phosphate and aniline (**1**), 2-hydroxyaniline (**2**), and 2-mercaptoaniline (**3**) were studied. Quantum chemical calculations revealed that the bathochromic shift in the UV-Vis spectrum of OH-substituted SB relative to the aniline derivative, and the hypsochromic shift in the UV-Vis spectrum of the thiol-bearing base are caused by a combination of two factors. The first one is the change in the dihedral angle value that leads to a decrease in the conjugation degree between two aromatic cycles in a row of **1** ≈ **2** < **3**. The second one is the contribution of oxygen or sulfur atoms to the frontier molecular orbitals involved in the electron transitions.

Hydrolysis resilience at different pH values also increases from the unsubstituted Schiff base to the 2-mercaptoaniline derivative through the 2-hydroxyaniline SB. This can be explained by the decrease in local nucleophilicity of the imine nitrogen according to rows **1** < **2** < **3**.

## Figures and Tables

**Figure 1 molecules-29-03504-f001:**
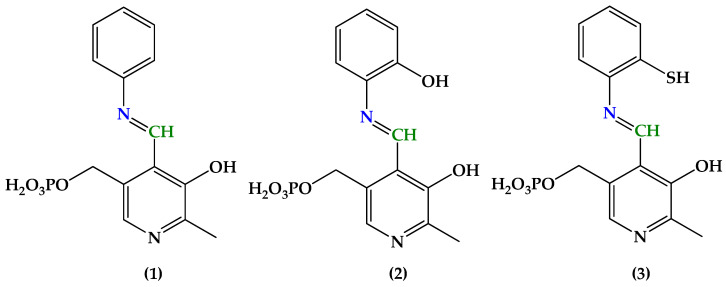
Structures of Schiff bases derived from pyridoxal 5′-phosphate and aniline (**1**), 2-hydroxyphenylamine (**2**), and 2-mercaptophenylamine (**3**). The key atoms involved in hydrolysis are marked with blue (–N=) and green (=CH–).

**Figure 2 molecules-29-03504-f002:**
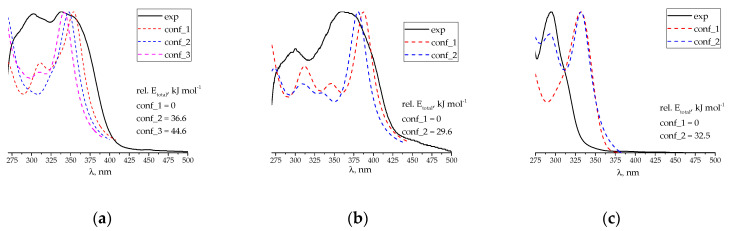
Normalized experimental (solid lines) and TD DFT-calculated (dashed lines) UV-Vis spectra of Schiff bases **1** (**a**), **2** (**b**), and **3** (**c**).

**Figure 3 molecules-29-03504-f003:**
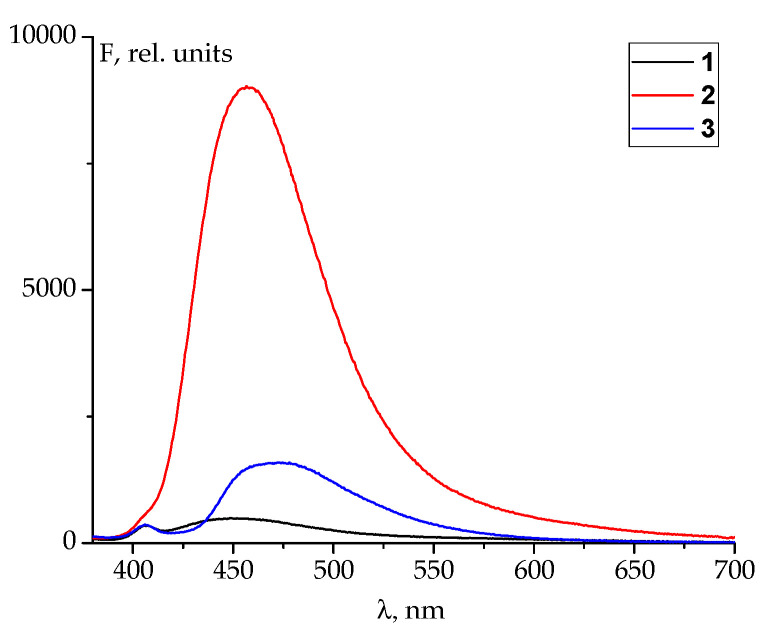
Fluorescent emission spectra of the Schiff bases derived from PLP and aniline (**1**), 2-hydroxyaniline (**2**), and 2-mercaptoaniline (**3**) registered using *λ_ex_* = 365 nm. SB concentration in DMSO is 10^−5^ mol L^−1^.

**Figure 4 molecules-29-03504-f004:**
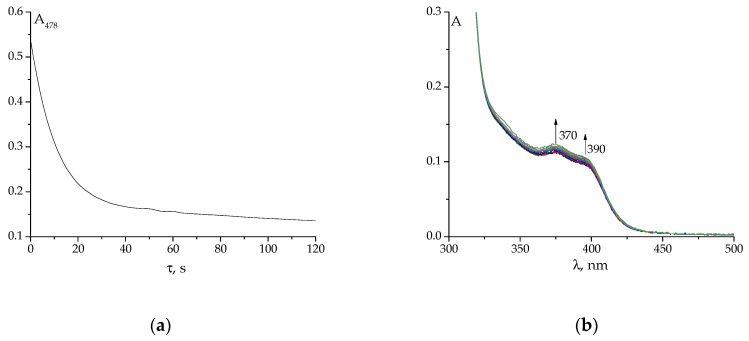
Changes in absorbance at λ = 478 nm of SB **2** (**a**) and evolution of UV-Vis spectra of SB **3** (20 spectra were registered with a delay of 45 s) (**b**) after quickly adding SBs to an aqueous buffer solution with a pH of 4.5. The final concentration of SB in the aqueous medium was 10^−4^ mol L^−1^.

**Figure 5 molecules-29-03504-f005:**
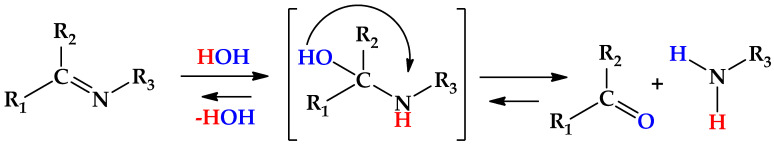
Schematic presentation of the mechanism of Schiff base hydrolysis (from left to the right) and formation (from right to the left).

**Table 1 molecules-29-03504-t001:** Selected vertical electronic transitions (UV–Vis absorption spectra) calculated using the TD DFT/B3LYP method for the different conformers of the Schiff bases derived from pyridoxal 5′-phosphate and aniline (**1**), 2-hydroxyaniline (**2**), and 2-mercaptoaniline (**3**).

Schiff Base/Conformer	Excited State	λ_cal_ (nm)	Oscillator Strength (*f*)	Composition *
**1**/1	A_1_	354.78	0.3492	HOMO→LUMO (100%)
	A_3_	309.68	0.1650	HOMO-3→LUMO (15%), HOMO-2→LUMO (68%), HOMO-1→LUMO (17%)
**1**/2	A_1_	347.97	0.3448	HOMO-1→LUMO (10%), HOMO→LUMO (88%)
	A_5_	267.44	0.3267	HOMO-5→LUMO (16%), HOMO-4→LUMO (79%)
**1**/3	A_1_	342.10	0.2732	HOMO→LUMO (93%)
	A_5_	258.10	0.3088	HOMO-5→LUMO (24%), HOMO-4→LUMO (72%)
**2**/1	A_1_	387.48	0.3186	HOMO→LUMO (100%)
	A_2_	345.96	0.0928	HOMO-1→LUMO (100%)
	A_4_	311.15	0.1442	HOMO-3→LUMO (55%), HOMO-2→LUMO (45%)
**2**/2	A_1_	380.57	0.3921	HOMO→LUMO (100%)
	A_2_	336.30	0.0830	HOMO-1→LUMO (100%)
	A_3_	315.36	0.0686	HOMO-3→LUMO (20%), HOMO-2→LUMO (78%)
**3**/1	A_1_	333.00	0.3150	HOMO-1→LUMO (12%), HOMO→LUMO (84%)
**3**/2	A_1_	332.70	0.2737	HOMO-2→LUMO (7%), HOMO→LUMO (86%)
	A_2_	294.26	0.1598	HOMO-2→LUMO (9%), HOMO-1→LUMO (91%)

* The transitions with a contribution less than 5% are omitted.

**Table 2 molecules-29-03504-t002:** Formation and hydrolysis rate constants and equilibrium constants of SBs **1**–**3** formation determined from the kinetic experiments.

Schiff base derived from PLP and aniline (**1**) *			
Parameter	pH = 4.5	pH = 7.4	pH = 9.5
log *k*_1_	2.34	3.25	3.04
log *k*_−1_	0.41	0.31	0.42
log *K*	1.93	2.94	2.62
Schiff base derived from PLP and 2-hydroxyaniline (**2**)			
Parameter	pH = 4.5	pH = 7.0	pH = 9.5
log *k*_1_	4.22 ± 0.10	4.11 ± 0.07	4.48 ± 0.01
log *k*_−1_	0.56 ± 0.10	0.41 ± 0.02	1.21 ± 0.05
log *K*	3.66 ± 0.14	3.70 ± 0.07	3.27 ± 0.05
Schiff base derived from PLP and 2-mercaptoaniline (**3**)			
log *k*_1_	3.50 ± 0.10	3.29 ± 0.08	2.51 ± 0.01
log *k*_−1_	−2.03 ± 0.10	−1.26 ± 0.07	−0.42 ± 0.01
log *K*	5.53 ± 0.14	4.55 ± 0.11	2.93 ± 0.01

* Data are adopted from the reference [39].

**Table 3 molecules-29-03504-t003:** Global electronic properties of the Schiff bases 1–3 and local electronic properties of their methine carbon and imine nitrogen (denoted with subscripts ‘C’ or ‘N’, respectively).

**Schiff Base**/Conformer	**1**/1	**2**/1	**3**/1
*µ*, eV	−4.50	−4.48	−4.54
*η*, eV	4.05	3.81	6.30
*ω*, eV	2.50	2.63	1.63
*N*, eV	2.55	2.69	2.89
*ω_C_*	−1.15	−1.23	−0.82
*ω_N_*	−0.37	−0.48	−0.22
*N_C_*	−0.72	−0.62	−0.77
*N_N_*	0.26	0.22	0.18
*R_C_*	−0.94	−0.93	−0.79
*R_N_*	−0.62	−0.69	−0.40

## Data Availability

Data is contained within the article or Appendix A.

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
