# Peer review of "Schiff Bases Derived from Pyridoxal 5′-Phosphate and 2-X-Phenylamine (X = H, OH, SH): Substituent Effects on UV-Vis Spectra and Hydrolysis Kinetics"

_molecules, 2024, doi:10.3390/molecules29153504_

Round 1

Reviewer 1 Report

Comments and Suggestions for Authors

Gamov and coworkers analyse the influence of OH and SH groups in the hydrolisis of Schiff Bases. Although the work is well-structured and the combination of both experimental and theoretical data is interesting, more experimental data with other substrates should be adquired to support this proposal and accept this manuscript for publication.

-        Authors explain the presence of H-bonds and the electronic donnor properties of -OH and -SH groups in their results. It would be interesting to analyze the influence of a OCH3 group at these positions instead (in each pyridine and phenyl moieties) and really understand the electronic nature of the imine. Are intermolecular H-bonds involved?

-        Moreover, it would be interesting to include the influence of both -OH and -SH groups at the para position of the phenyl ring.

-    On the other hand, I am not completely sure if Mulliken charges are consistant enough to support the electronic properties of the compounds. Is there any other more appropiate method? In addition, is there any previous bibliography about the change of the diedral angle promoted by OH or SH? Please mention.

Minor changes:

-         Please draw the different proposed isomers to facilitate further discussion

-          Line 152: “Can the hydrolysis resilience be also different? The next Section answers this question”. Change this sentence by: “ the hydrolisis resilience will be evaluated in the following section”

-          Line 180: general reaction instead of formula.

Author Response

Dear Reviewer,

please, find the detailed response to your comments enclosed.

Reviewer 2 Report

Comments and Suggestions for Authors

This work is dedicated to indicating that if the OH- and SH- groups introduced into the phenyl ring can enhance the stability of Schiff bases (SBs). SBs are recognized as versatile compounds, but their susceptibility to hydrolysis in an aqueous medium limits the application of most SBs. Improving the stability of SBs in aqueous medium is beneficial for broadening the range of applications of SBs. In this manuscript, the authors investigated the substituent effect on the UV-Vis spectra through both experimental and calculated spectra. Additionally, the hydrolysis (k-1) and formation (k1) rate constants for SB 1, 2 and 3 at different pH values were calculated based on experimental data. The experimental data revealed that the OH- and SH- groups introduced into the phenyl ring enhance the stability of SBs, global and local electronic properties of SB 1, 2 and 3 were calculated to elucidate the stability. Overall, it appears that the authors aimed to offer suggestions for synthesizing SBs that can be utilized in aqueous medium, but further work is needed before publication.

1.       The authors should explain how to obtain the conformers used to calculate the UV-Vis spectra and global and local electronic properties. For example, if the “scan” in Gaussian09 was utilized, authors should provide the coordination variables(CVs) they used and the corresponding energy along the CVs.

2.       I want to know why UV-Vis spectra were provided in this manuscript. In my view, the authors aimed to investigate how to increase the stability of SBs, but there is no information provided from UV-Vis spectra to enhance the stability of SBs.

3.       As shown in Fig. 5, the mechanism of the Schiff base hydrolysis was reversible. The formation and hydrolysis of SBs both influence the existence of SBs in aqueous medium. However, in this manuscript, the authors only provided the global and local electronic properties of SBs. It means that only the hydrolysis of SBs was considered. Therefore, it would be beneficial to provide the global and local electronic properties of the product molecules.

Comments on the Quality of English Language

Please try to set the target of this manuscript more clearly. Authors should describe the significance of their results to provide proper suggestions for enhancing the application of most SBs in an aqueous medium.

Author Response

(The authors gave the same response as above.)

Round 2

Reviewer 1 Report

Comments and Suggestions for Authors

Thank to the authors for the changes. I think these changes have improved the quality of the manuscript, but more data should be included before acceptance for publication: 

- Please include all the structures of the molecules with their respective number in Figure 1. It is better for the reader to understand the manuscript

- And more important, please include hydrolisis analysis for the new compounds (OMe derivatives and the para sustituted ones) in both results and conclusion parts. These new compounds should be useful not only to analyze the hydrogen bonding, but also to complete the hydrolisis study and stablish a general pattern between all the derivatives.

Author Response

Dear reviewer,

once again, thanks for the attention paid to our manuscript and quick response. Please, find our answers below:

  1. Q.: "- Please include all the structures of the molecules with their respective number in Figure 1. It is better for the reader to understand the manuscript"
    A.: According to your comment, a separate Figure is created (Fig. S1), which contains the structures of all conformers in one place. The necessary explanations, of how they can be obtained one from another are added to the Fig. S1 caption.
  2. Q.: "- And more important, please include hydrolisis analysis for the new compounds (OMe derivatives and the para sustituted ones) in both results and conclusion parts. These new compounds should be useful not only to analyze the hydrogen bonding, but also to complete the hydrolisis study and stablish a general pattern between all the derivatives."
    A.: We appreciate your insightful suggestion regarding the inclusion of additional data. We completely agree that such information would enhance our findings. However, unfortunately, the synthesis of Schiff bases containing methoxy groups is still pending. Procuring the necessary reagents may take several months, which delays our ability to produce these compounds.
    Moreover, we believe that exploring the hydrolysis of the methoxy-substituted Schiff bases may extend beyond the scope of our current study. Our primary focus has been on Schiff bases 2 and 3, which have already demonstrated their utility as fluorescent sensors for albumins (10.1016/j.ymeth.2022.08.014, 10.1016/j.jphotochem.2023.114905). Therefore, we concentrated our efforts on these specific compounds.
    We are committed to further investigations into other structural analogs of the Schiff bases in future research endeavors.

Best regards,
On behalf of all co-Authors,

George Gamov,

Ivanovo State University of Chemistry and Technology